# A Review of the Giant Triton (*Charonia tritonis*), from Exploitation to Coral Reef Protector?

Cherie A. Motti, Scott F. Cummins and Michael R. Hall

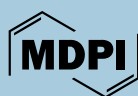

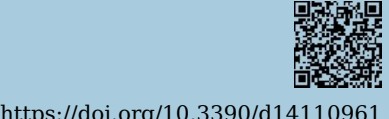

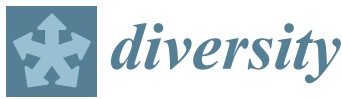

# A Review of the Giant Triton (*Charonia tritonis*), from Exploitation to Coral Reef Protector?

Cherie A. Motti [1,*], Scott F. Cummins [2,3] and Michael R. Hall [1]

1 Australian Institute of Marine Science (AIMS), Townsville, QLD 4810, Australia
2 Centre for Bioinnovation, University of the Sunshine Coast, Maroochydore, QLD 4558, Australia
3 School of Science, Technology and Engineering, University of the Sunshine Coast, Maroochydore, QLD 4558, Australia
* Correspondence: c.motti@aims.gov.au; Tel.: +61-7475-34143

**Abstract:** *Charonia tritonis* (Charoniidae), one of the largest marine gastropods and an echinoderm specialist, preys on Crown-of-Thorns starfish (CoTS), a recurring pest that continues to be a leading cause of coral mortality on Indo-Pacific reefs. Widespread historical exploitation has impacted their numbers, with standing populations considered rare throughout their habitat. Their life-stage attributes, i.e., teleplanic larvae, planktotrophic phase spanning years permitting transoceanic dispersal, and recruitment to coral reefs through oceanic influx with intense larval mortality, have likely hindered their recovery. Decline in numbers is hypothesised to account partially for periodic CoTS outbreaks, yet predator-prey dynamics between these two species that might influence this are poorly understood. The *C. tritonis* excretory secretome elicits a proximity deterrence effect on CoTS, the nature of which is under investigation as a possible tool in CoTS biocontrol scenarios. However, specificity and zone of impact in situ are unknown, and whether the mere presence of *C. tritonis* and/or predation pressure has any regulatory influence over CoTS populations remains to be established. The fundamental taxonomy and distinctive characteristics, biology and ecology of *C. tritonis* is summarized, and knowledge gaps relevant to understanding their role within coral reefs identified. Information is provided regarding exploitation of *C. tritonis* across its habitat, and prospects for conservation interventions, including captive rearing and stock enhancement to repopulate local regions, are discussed. Its predator-prey ecology is also examined and potential to mitigate CoTS considered. Recommendations to direct future research into this predator and for its inclusion in a CoTS integrated pest management strategy to improve coral reef health are offered.

**Keywords:** aquaculture; biocontrol; Crown-of-Thorns starfish; indigenous predator; integrated pest management; marine gastropod; trophodynamics; predation efficiency

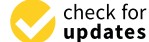



## 1. Introduction

*Charonia tritonis*, commonly known as the giant triton, is one of the largest marine gastropod snails. The historical (and continuing) exploitation of the *C. tritonis* as a curio throughout its full habitat range has led to a significant decline in its numbers, to the point of it becoming rare and endangered, and localised protection has not alleviated the problem [1,2]. As an echinoderm specialist and the primary predator of adult Crown-of-Thorns starfish (CoTS; *Acanthaster* cf. *solaris* species complex [3,4]), its deliberate removal may have also altered trophic interactions, triggering cascading effects on coral reef ecosystem processes [5]. Endean's [6] 'predator removal hypothesis' proposed the decline in the *C. tritonis* population as a possible driver for the alarming increase in CoTS numbers (referred to as outbreaks), a concept that is still being debated several decades later [4,6–9]. The increase in CoTS outbreak frequency and severity continues to have a devastating impact on coral reef ecosystems, prompting a resurgence in research and management efforts to control CoTS numbers [10–12]. A compounding factor is the elevated reproductive success

of CoTS and high larval survival rates, the latter augmented by the same stressors that adversely impact coral reefs, further fueling the need to develop novel CoTS management adaptive tools and methods [13–15]. In contrast, *C. tritonis* is relatively understudied, and its role in coral reef trophodynamics is poorly understood [12].

In recent years, biocontrol methods as a strategy to mitigate CoTS outbreaks (i.e., ecological goal to reduce CoTS populations to levels below damaging thresholds) on a regional scale have gained traction [16,17]. The success of biocontrol programs relies on efficient selection of effective natural enemies [18–24], and to achieve this, it is critical to have full knowledge of the pest's biology, as well as that of their natural enemies. Additionally, in any biocontrol effort, conservation of natural enemies is a critical component and requires knowledge not only of the predator's effectiveness against the pest species, but also the factors which interfere with or threaten their natural populations. As a natural indigenous enemy, the potential restocking of *C. tritonis* as a conservation intervention may also present the opportunity to naturally control CoTS populations on selected 'at-risk' reefs. The inclusion of such a strategy within a considered and complementary multi-faceted CoTS integrated pest management (IPM) program is of interest to reef managers [17,25,26], however, there exists many clear and evident knowledge gaps, least of all in the breeding and rearing of juvenile *C. tritonis* [27] for stock enhancement.

Presented here is a comprehensive review of literature on the *C. tritonis* taxonomy and morphology, biogeographical distribution, movement ecology, reproduction and growth. The state of exploitation and the anthropogenic threats they face are evaluated, and prospects for their captive rearing and restocking on the Great Barrier Reef (GBR; Australia) is discussed, the intent being to assist in the effective management and protection of their populations. Attributes suited to their use as a biocontrol agent to mitigate CoTS population outbreaks naturally and sustainably in the long-term, are also examined in the context of predator-prey dynamics, and recommendations to guide future research and establish environmental management strategies, with respect to their application within an IPM approach, are offered.

## 2. Taxonomy and Distinctive Characteristics

Members of Hypsogastropoda (phylum: Mollusca, class: Gastropoda, subclass: Caenogastropoda) are numerically important key predators in shallow water tropical marine environments [28,29]. Within clade Caenogastropoda, the Hypsogastropoda comprises the Non-Latrogastropoda clade corresponding largely to the former Littorinimorpha, with radula typically having 7 teeth per row, and the Latrogastropoda clade, which includes the previously named Neogastropoda clade and the Calyptraeoidea, Cypraeoidea, Ficoidea, Stromboidea, Tonnoidea, and Xenophoroidea, with radula having only 1–5 teeth per row [30]. The largest shells are associated with the Latrogastropoda superfamilies Turbinellidae (*Syrinx aruanus*) and Tonnoidea, the latter having been recently updated, based on mitochondrial and nuclear gene analysis [31], to comprise nine families: Bursidae, Cassidae, Charoniidae, Cymatiidae, Laubierinidae, Personidae, Ranellidae, Thalassocyonidae and Tonnidae. Some species within the Tonnidae, Cassidae, Cymatiidae and Charoniidae families are known to prey on echinoderms, for example: *Tonna perdix*, *T. galea* [32] and *T. zonatum* [33]; *Cassis tuberosa* [34] and *Galeodea echinophora* [35]; and *Charonia* spp. [36–38], respectively [31]. *Charonia lampas* (Linnaeus, 1758) [30,39], previously classified into five subspecies (*C. lampas capax*, *C. lampas lampas*, *C. lampas pustulata*, *C. lampas rubicunda* and *C. lampas sauliae*), is the most morphologically variable of the *Charonia* genus, driven by ecophenotrypic rather than genetic variation [40,41] (Table 1). *Charonia seguenzae*, having been geographically isolated in the Eastern Mediterranean Sea and therefore split from *C. variegata* (Lamarck, 1816) [42], has since been reclassified as *C. variegata*. *C. tritonis* (Linnaeus, 1758) is the largest species within the genus and the only *Charonia* known to predate on CoTS.

**Table 1.** Taxonomic status of the genus *Charonia*. Key taxonomic groups in bold, * denotes initial subspecies. Adapted from [30,31,39].

| Scheme | Author(s) | Status | Accepted Name |
|---|---|---|---|
| *Charonia lampas* | **Linnaeus, 1758** | **accepted name** | *Charonia lampas* |
| *Charonia lampas capax* * | Finlay, 1926 | synonym | *Charonia lampas* |
| *Charonia capax euclioides* | Finlay, 1926 | synonym | *Charonia lampas* |
| *Charonia crassa* | Grateloup, 1847 | synonym | *Charonia lampas* |
| *Charonia euclia* | Hedley, 1914 | synonym | *Charonia lampas* |
| *Charonia euclia instructa* | Iredale, 1929 | synonym | *Charonia lampas* |
| *Charonia lampas lampas* * | Linnaeus, 1758 | synonym | *Charonia lampas* |
| *Charonia lampas macilenta* | Kuroda & Habe, 1961 | synonym | *Charonia lampas* |
| *Charonia (lampas) pustulata* * | Euthyme, 1889 | synonym | *Charonia lampas* |
| *Charonia lampas sauliae* * | Reeve, 1844 | synonym | *Charonia lampas* |
| *Charonia lampas weisbordi* | Gibson-Smith, 1976 | synonym | *Charonia lampas* |
| *Charonia lampas ventricose* | Grateloup, 1833 | Synonym | *Charonia lampas* |
| *Charonia mirabilis* | Parenzan, 1970 | synonym | *Charonia lampas* |
| *Charonia nodifera* | Lamarck, 1822 | synonym | *Charonia lampas* |
| *Charonia lampas rubicunda* * | Perry, 1811 | synonym | *Charonia lampas* |
| *Charonia powelli* | Cotton, 1956 | synonym | *Charonia lampas* |
| *Charonia tritonis* | **Linnaeus, 1758** | **accepted name** | *Charonia tritonis* |
| *Charonia variegata* | **Lamarck, 1816** | **accepted name** | *Charonia variegata* |
| *Charonia variegata seguenzae* * | **Aradas & Benoit, 1870** | **accepted name** | *Charonia variegata* |
| *Charonia tritonis variegata* | Lamarck, 1816 | synonym | *Charonia variegata* |

An extraordinary adaptive radiation driven through diet and competition has seen the morphological, physiological, behavioural and ecological diversity of the Hypsogastropoda expand [43–45]. Several apomorphic forms (or derived traits) within this clade of caenogastropods, and which are shared by *Charonia* spp., predominantly relate to the digestive system, specifically: a distinct rectal gland, salivary glands that do not pass through the nerve ring, tubular accessory salivary glands, possession of a 1–5 toothed radula, an esophageal gland detached from the esophagus, an enlarged radula ventral tensor muscle to aid sliding [28,45,46], formation of an eversible proboscis, a specialised siphon, and repeated folding in the chemoreceptor osphradium to increase the surface area capable of selective and acute chemical sensitivity [47]. Furthermore, they differ from herbivorous gastropods, having complex behaviour adaptations that include searching, capture, immobilization and penetration of prey, as well as an altered biochemical composition suited to the digestion of animal tissues [47,48].

*Charonia* spp. are readily distinctive, having a large, tall shell measuring up to 500 mm in length, with a pointed spire. The shell consists of a large body whorl with broad cords within a single narrow thread filling each interspace and a pronounced flaring outer lip. Well-developed varices are evenly spaced approximately every 270° around the shell, each merging abaperturally and bearing the remnants of older outer lips. The shell is high gloss with a contrasting colour pattern of red-brown crescentic splashes against a cream to pink background [39]. There is no periostracum. The brown oval shaped operculum is distinct in that it has concentric growth lines. The radula is the most distinctive non-shell feature, consisting of a central broad tooth with a very narrow basal plate that curves downward at the extremities and is surrounded by several narrow, elongated teeth [39]. Prominent features of the *C. tritonis* shell include the presence of smooth, broad, and flattened spiral ribs, the edge of which are wavy and puckered, and a broad, short siphonal canal with thin folds along the columellar wall.

## 3. Distribution, Habitat and Abundance

*Charonia* spp. have an extensive geographical distribution (Figure 1A). *C. lampas* ranges from the temperate waters of the Atlantic [49] to the sub-tropical waters of the Indian and Pacific Oceans [41,50] (Figure 1A). It is commonly found in the western Mediterranean, but

largely absent in the eastern Mediterranean [42,50]. The species has been documented in the northern coast of Natal in South Africa, southern and eastern Australia [51,52], New Zealand, the Chatham, Kermadec, Raoul, Norfolk and Lowe Howe Islands and around the islands of Japan and Taiwan [53]. *C. variegata* (Lamarck, 1816) is distributed through the western tropical Atlantic, the Caribbean and subtropical regions of the Mediterranean [42,50,54,55] (Figure 1A). As a result of the late Pliocene uplift of the Isthmus of Panama *C. variegata* has been geographically, and hence genetically, separated from *C. tritonis*.

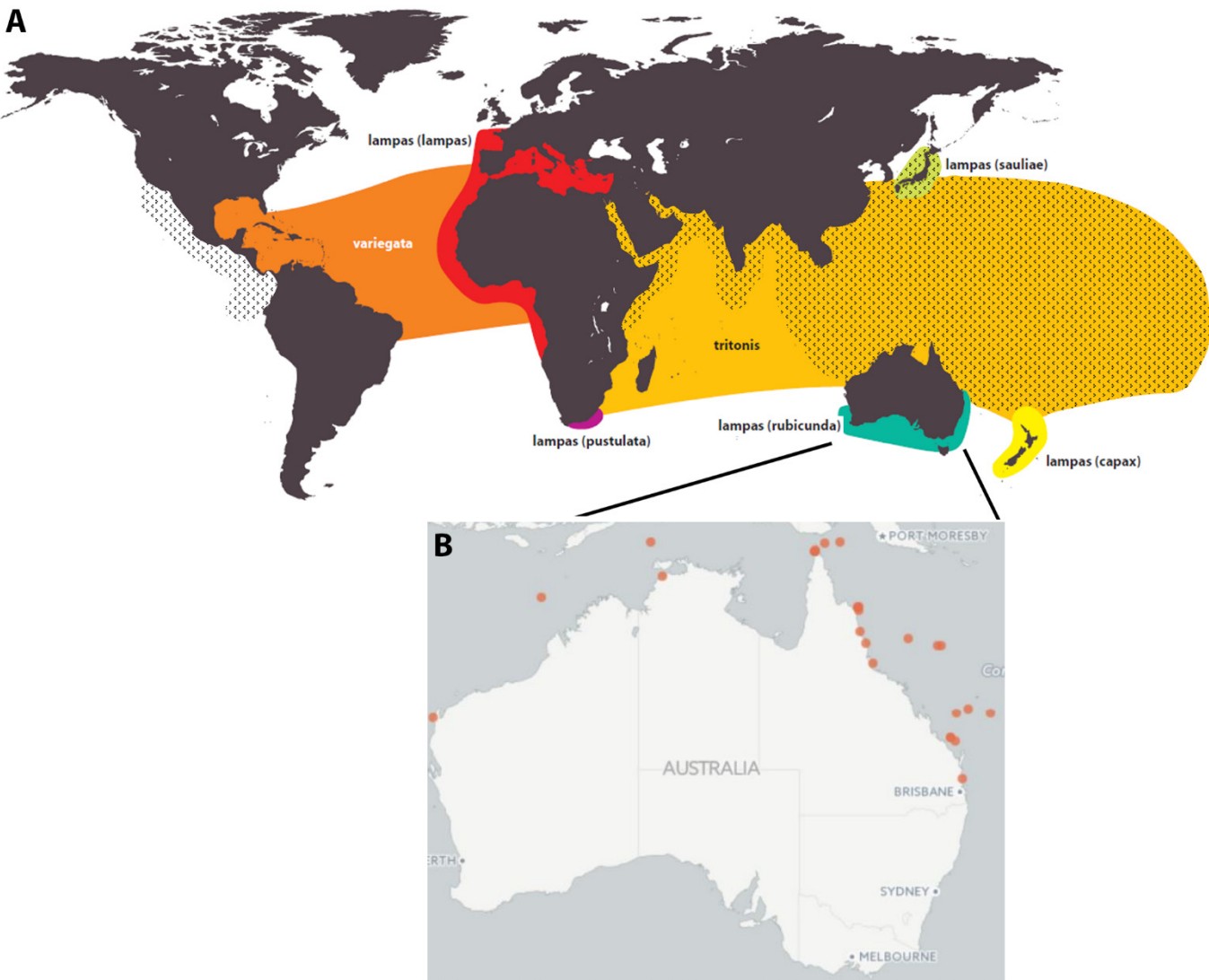

**Figure 1.** Distribution and range of (**A**) the *Charonia* genus, coloured by species, and *Acanthaster* cf. *solaris* [10], shown by hashed overlay, and (**B**) locations of reported sightings and collections of *Charonia tritonis* in Australia, shown by orange dots. Adapted from [56–58].

Of all the *Charonia* spp., *C. tritonis* has the greatest distribution, extending throughout the tropical Indo-West Pacific region [57,58] (Figure 1A). It has been documented from central Japan [59], tropical Australia (Figure 1B), New Zealand and the Pitcairn, Cocos, Galapagos, Easter and Hawaiian Islands [50,60,61]. Its range also extends from the Red Sea to southern East Africa and across the islands of the Indian Ocean to western Australia [52].

*Charonia tritonis* inhabits hard and sandy bottoms in and around shallow water coral reefs [62,63], although some specimens have been observed at depths of several hundred meters [39]. They are generally considered nocturnal, and their cryptic nature (hiding in crevices during the day) makes accurate sampling to survey population size non-trivial [64].

With the full-length mitochondrial genome of *C. lampas* [65] and *C. tritonis* sequenced [66], the identification of candidate sequences suitable for use as species-specific barcodes in environmental DNA (eDNA) technology [67,68], in combination with impromptu citizen surveys, may assist in establishing their current spatial distribution and true numbers. Accurate mapping of their spatial co-occurrence with CoTS (Figure 1A) will also provide some insight into their potential role in CoTS outbreaks and biocontrol.

## 4. Movement Ecology

Understanding movement ecology (e.g., foraging, dispersal and seasonal migration) has proven critical to the management and conservation of several marine species [69]. Importantly, such information underpins a species' population distribution, which is also influenced by seasonal phenology and predator-prey interactions [70,71]. Yet, information is scarce regarding the habitat and home range of the *C. tritonis*, their fine-scale movements as they move through their habitat, their normal (non-stressed) behaviours and how they interact with prey species (including CoTS) in situ. Recently, modelling of acoustic array data found *C. tritonis* have the capacity to move across an entire local reef [72] and are likely to be able to move between adjacent reef systems, having been found at depth on sandy bottoms [39]. Conversely, acoustic tagging of CoTS revealed they did not move beyond a single receiver within a linear array over a four month deployment, moving less than 100 m [73]. If these results prove robust, the home range of *C. tritonis* is well beyond that of their CoTS prey, at least on reefs in non-outbreak status. The small home range of CoTS is likely linked to food availability, sedentary coral prey and limited predation pressure, whereas the larger home range of *C. tritonis* is more likely a consequence of their low population, mobile and cryptic prey (not just CoTS), and the need to locate a mate. Preliminary findings [16,74] have shown the CoTS excretory secretome acts as an attractant to *C. tritonis* and that CoTS exhibit a flight response to *C. tritonis* predator odor. Yet, revealing the nature and role of specific excretory semiochemicals (e.g., kairomones and pheromones) in motivating both predator and prey behaviors [75] is required to establish the parameters that describe the full range of each animal's behaviour and movements, as well as their predator-prey dynamics, in the field.

The movement rates of *C. tritonis* and CoTS need to be considered in tandem to establish the extent to which the predator alters the behaviour of CoTS and exerts downward pressure on the population. Based on modelling of acoustic tagging data, release of *C. tritonis* on a local reef has the potential to alter CoTS behaviour in the short-term [72], possibly forcing them to become more cryptic and forcing them take greater risk. However, it remains to be seen whether increasing *C. tritonis* populations (via conservation or restocking) over the longer-term will impact local or regional CoTS populations.

## 5. Reproduction

Members of the Tonnoidea are always gonochoric. Female *C. tritonis* can pair with multiple males during a single copulation event, the pairing lasting for several hours [76]. Copulation in captivity has been observed from August to September (Yongxing Island, China) with egg laying approximately 130 days after [76] (Table 2). A similar gestation period has been observed in other northern Pacific *Charonia* species [77], although much shorter time periods of 30–60 and 90 days have been reported [78,79]. No seasonality was observed in the reproduction of captive *C. tritonis* held for over two years at the Phuket Marine Biological Station, Thailand [78,80], indicating the photoperiod is not necessarily the cue for spawning. Furthermore, several species of Cymatiidae, held in captivity under ambient conditions (i.e., without controlled lighting), were observed spawning at precisely the same time over three consecutive years [81], giving credence to the suggestion that water temperature is a primary determinate [82]. This has since been confirmed for *C. tritonis* [25].

Detailed observations of *C. tritonis* reproduction, including copulation, spawning, embryogenesis and hatching, have been described in detail [25,78,79,83]. Internal fertilization represents a major innovation in the Caenogastropoda, along with encapsulation of the

eggs, both of which provide a protected environment during early trochophore development [28]. Briefly, the male sperm and prostatic fluid is transferred during copulation via the penis located behind the right tentacle [81]. Fertilization takes place internally in the female [55,84,85], the inseminated sperm embedding in the non-ciliated nutrient-rich surface cells of the pallial oviduct [81]. Under the right conditions (~26 °C water temperature; April to June coinciding with the austral winter solstice photoperiod and the dry season [25]) mature eggs are discharged into the pallial oviduct where they are fertilized, producing between 2000–2750 heavily yolked orange-coloured embryos ~360–600 μm in diameter (Figure 2A) (Table 2). Batches of embryos are deposited into ootheca (oblong tear-dropped shaped gelatinous capsules ~34–60 mm long and ~10 mm diameter), each containing a clear albuminous fluid. As each individual ootheca passes from the oviduct it is cemented via one end to a vertical (often cryptic) rocky surface [55,78,80]. The outer layer hardens upon exposure to the seawater offering protection against biofouling, pathogens and predation. The egg mass of 50–1000 oothecae, containing up to $1.47 \times 10^6$ eggs (Table 2) [78,80] may take up to a week to deposit [81,86]. At the end of the breeding season both the testis and ovary degenerate [81].

**Table 2.** Reproductive statistics for *Charonia tritonis* reported from ex situ breeding programs. CoTS = Crown-of-Thorns starfish; PSU = practical salinity unit.

| | Berg (1971) [79] | Nugranad et al. (2000) [80] | Nugranad et al. (2001) [78] | Zhang et al. (2013) [76] | Motti et al. (2019) [25] |
|---|---|---|---|---|---|
| Location | Oahu, USA | Phuket, Thailand | Phuket, Thailand | Yongxing Island, China | Townsville, Australia |
| Number of females | 1 | 1 | 5 | 2 | 4 |
| Number of males | At least 1 | - | At least 1 | At least 1 | At least 2 |
| Broodstock diet | Natural diet | CoTS, Culcita novaeguineae, Holothuria atra and Stichopus chloronotus | CoTS, C. novaeguineae, H. atra and S. chloronotus | CoTS and Stichopus horrens | CoTS, Linckia sp. and S. chloronotus |
| Date of reproductive behaviour | Oct | - | Year round | August–September | March-June |
| Temperature of broodstock tank | - | 25.5–33.0 °C | - | - | 23 °C (winter)–30 °C (summer) |
| Copulation until laying (days) | 120–150 | - | 30–60 | 133 | - |
| Duration of spawning (days) | 42–56 | 19 | 60 | 21–35 | - |
| Temperature of egg hatchery | - | - | - | 24 °C | 24.5 °C |
| Total capsules spawned female$^{-1}$ | 88+ | 50 | 500–1000 | 549–602 | ~400 |
| Egg diameter (μm) | 450–600 | 400–430 | 360–440 | 428 | - |
| Capsule dimensions, H × L (mm) | 25 H × 9 L | 17–39 H × 9–10 L | 17–39 H × 9–10 L | 34 H × 9 L | - |
| Number of eggs per capsule | - | 2000–3400 | 2000–4400 | 2740–3000 | ~2500 |
| Total number of eggs produced | - | $~1.5 \times 10^5$ | $1.6 \times 10^6$–$3.2 \times 10^6$ | $1.5 \times 10^6$–$1.6 \times 10^6$ | - |

**Table 2.** *Cont.*

| | | | | | |
|---|---|---|---|---|---|
| Incubation period (days) | 49–56 | - | 35–60 | 55–63 | 52–68 |
| Hatching success of capsules | - | 0% unfertilized | 43–96% | 86–96% | - |
| Veligers per capsule | 1140–1447 | - | 973–1459 | 2046–2110 | - |
| Total veligers produced female$^{-1}$ | - | - | $0.26 \times 10^6$–$1.47 \times 10^6$ | $1.12 \times 10^6$–$1.27 \times 10^6$ | ~$0.8 \times 10^6$ |
| Shell length at hatching (µm) | 768–934 | - | 720–925 | 664–700 | 740 |
| Temperature of larval rearing tank | - | - | - | - | 24.5 °C |
| Larval diet | - | - | - | Immediately post hatching: *Isochrysis zhanjiangensis, Chaetoceros muelleri* and *Phaeodactylum tricornutum* (1:1:1, 2.0–3.0 $\times 10^4$ cells mL$^{-1}$). Two weeks post hatching: formulated brine shrimp flakes (52% protein, 8% crude fat, 5% crude fiber, and 7% moisture) at a rate of 0.3 mg L$^{-1}$ every other day. | *Isochrysis galbana, Diacronema lutheri, Nannochloropsis oceania, Dunaliella* sp. |
| Other conditions | | 32–34 PSU | | | 34–36 PSU 0.7 veliger mL$^{-1}$ |
| Settlement | None at 30 days | - | None at 300 days | None at 140 days | None at 83 days |

All eggs within the ootheca have an equal chance of undergoing embryogenesis as there are no nurse cells [81]. This post-fertilization process occurs within the egg capsule and can take anywhere from 35 to over 60 days [25,76,78,79]. Throughout this incubation period the female does not feed. She exhibits maternal care, using her foot muscle to clean the outer surface of the egg capsule, thereby preventing biofouling [76,87], and physically protect the oothecae from predation [81,88–90]. Other females in the vicinity may also protect and care for the clutch [81,87].

*C. tritonis* embryos gastrulate at ~7 days post-fertilization (dpf). Trochophore development occurs between 9–12 dpf followed by protoconch I (or embryonic shell) formation at 15–18 dpf [76]. This development phase is typified by the formation of the first two shell whorls. As yolk reserves are depleted, there is an overall whitening of the egg capsules, their texture becoming granular. The two eyespots develop 25 dpf (Figure 2B), the operculum and foot begin to form at 29 dpf, and a larval (false) heartbeat can be detected at ~35 dpf [76]. After ~63 dpf, trochophores, having a shell length of between 664–934 µm, emerge from the ootheca through a terminal pore [81] and enter their planktotrophic phase.

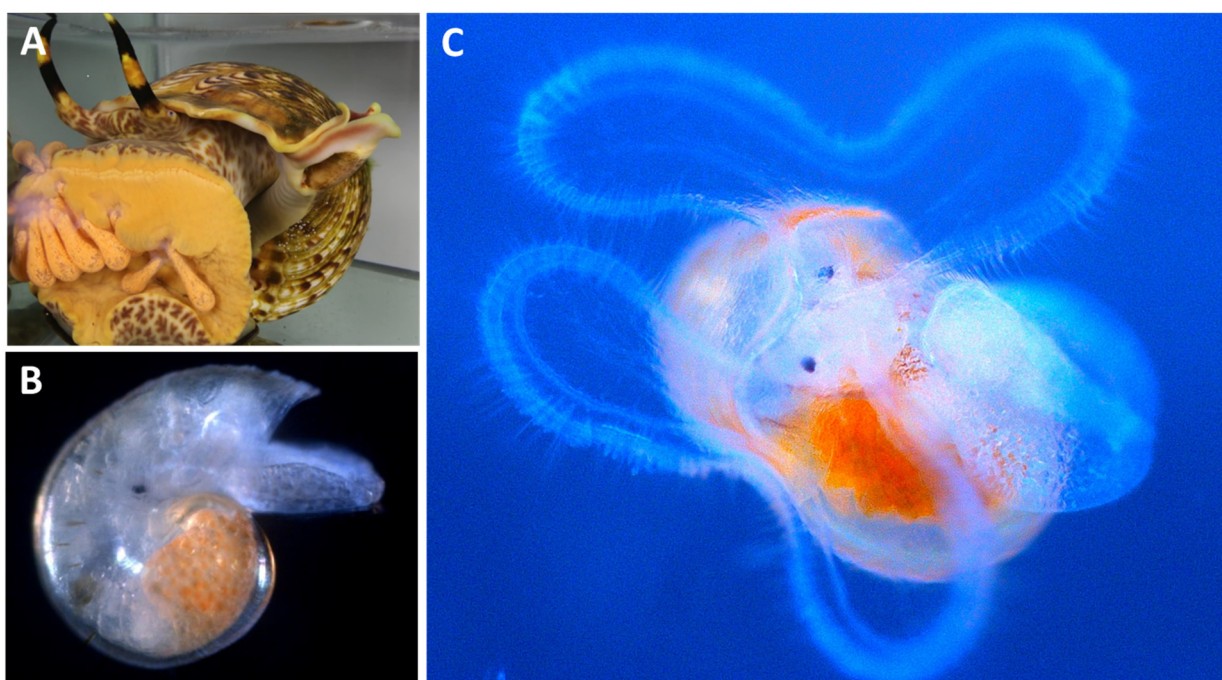

**Figure 2.** The early life history stages of *Charonia tritonis*. (**A**) Adult female depositing egg capsules in the SeaSim laboratory at AIMS (Photo: Peter Thomas-Hall, AIMS), (**B**) side profile of 25-day post fertilization veliger showing yolk reserve (orange) and eye spot (black) (Photo: Tom Barker, AIMS) and (**C**) front view of 14-day post hatched veliger with two eye spots visible and velum extended (Photo: Thomas Armstrong, AIMS).

In captivity, a female *C. tritonis* has been observed to produce $0.26–1.47 \times 10^6$ veligers per spawning season (Table 2). This level of fecundity, assuming it holds true for wild breeding females, raises the question: why are *C. tritonis* rare in locations where they have been actively protected by policy? The relative population densities on coral reefs of adult gastropod species which have planktonic and planktotrophic larval development compared to those with intracapsular development, i.e., Cypraeidae (pelagic phase 10–50 days) versus Volutidae, is reported to be 1:10 [91]. Therefore, larval survival in *Charonia* spp. is likely to be extremely low and/or their dispersal so great that settlement to any one reef, especially the natal spawning reef, is rare [92], possibly hindering population recovery [93].

A lack of understanding of the rudimentary requirements of the *C. tritonis* veliger, including information regarding the biochemical and physiological processes that regulate adult reproduction, larval development and larval growth [25,94], have hampered attempts to rear them in captivity. A recent de novo assembly of transcriptomes from the *C. tritonis* cerebral ganglion identified 38 neuropeptide precursor genes encoding for conserved molluscan neuropeptides, including several associated with reproduction [74]. Future studies with a focus on de novo whole-genome sequencing of the *C. tritonis* genome and additional transcriptomic studies targeting the functional characterisation of these conserved molluscan neuropeptides are needed to decode the *C. tritonis* reproductive neuroendocrine pathway [95] and better understand their social behaviour. In addition, complementary genetic studies to maximize egg and embryo viability and survival of veliger and juveniles are needed ([96] and references therein). For example, genomic estimated breeding values established based on genome-wide single nucleotide polymorphisms and growth traits (i.e., shell length, height, width, and weight, and body weight), has imparted significant growth advantages to offspring of the commercially important freshwater triangle sail mussel, *Hyriopsis cumingii* [97]. Such knowledge of *C. tritonis* will promote the development of more reliable aquaculture methods to support stock enhancement and will be especially important should it prove an important conservation biocontrol agent against CoTS [98–101].

## 6. Juvenile Growth, Development and Morphological Relationships

The veliger of Ranellidae, Cymatidae and Charoniidae, including *C. tritonis*, hatch at an advanced protoconch stage I and complete their development in the water column [102]. They are teleplanic, having an extraordinarily long larval development phase capable of dispersing across oceans [103–105]. For example, the larval duration of *Fusitriton oregonensis* (Cymatiidae), from hatching to metamorphosis, can extend up to 4.5 years, the longest teleplanic larval period recorded for any marine organism [106].

The shell length of newly hatched *C. lampas* veliger measures 430 μm. For *C. variegata* the shell measures between 770–930 μm [79,86], with some specimens collected from the Atlantic exceeding 5000 μm [105]. Shell length of *C. tritonis* veligers maintained for over 140 days [76], 164 days [25,107] (Figure 2C) and 300 days [80] all progressively increased over time, reaching approximately 2000 μm. However, although shedding of the velar cilia was observed, none achieved the protoconch II phase [90,104] or successfully advanced to settlement [78], and factors such as the minimum shell size required for the transition remain unknown.

Attainment of settlement competency relies on numerous factors including a minimum shell size, sufficient energy reserves and the development of specific receptors and neural connections [108,109]. Even once competency is achieved these veliger can halt growth and limit calcification enabling them to remain planktonic and, endowed with four large velar lobes extending up to 10 times the diameter of the larval shell length, transit oceans [103,105,110–113] presumably until they encounter a suitable and often highly specific settlement cue [114–119]. High density monocultures of various commercially important molluscan bivalves, i.e., oysters, clams, scallops, etc., can be induced to settle with high success when presented with various substrates and/or environmental chemical cues (Table 3). Late development stage teleplanic Tonnoidea larvae captured in ocean plankton tows have metamorphosed and settled in aquaria, with biofilms on the tank walls speculated to be the source of the settlement signal [103,110]. Some gastropod juveniles start as ectoparasites and there is direct evidence larval settlement in these species is induced by waterborne cues from their adult (mostly sedentary algae, sponge and coral) prey [120], although for Tonnoidea veligers there remains only indirect evidence. For example, larvae of *Monoplex* (*M. aquatilis*, *M. nicobaricus*, and *M. pilearis*) and *Gutturnium muricinum* (both previously *Cymatium*) will settle in the presence of adult tridacnid clam prey [121]. Unidentified juvenile gastropods, speculated to be those of *C. tritonis*, have been reported to settle and parasitize starfish, especially *Echinaster lozonicus* and *Linckia multifora* [64]. Overall, there is only indirect evidence that Tonnoidea veligers rely on the odor of their future prey as a settlement cue [122]. For *C. tritonis* the cues that induce settlement remain elusive [25] (Table 3).

The application of -omics techniques has identified the molecular mechanisms and settlement cues for a range of gastropod veliger [95]. Recently, reference *C. tritonis* transcriptomes have been derived from adult tissues [37,74,148] and early life developmental stages (embryo and veliger) [101]. A diversity of rhodopsin-like G protein-coupled receptors (GPCRs), all representing candidate olfactory receptors, were located within adult cephalic tentacles, supporting earlier studies showing *C. tritonis* use chemosensing to locate CoTS prey [16,37]. In addition, several GPCR genes were identified as being unique to veligers providing insight into the chemosensory capacity of this early life stage with possible function in settlement. While such findings are beginning to address the knowledge gaps, further investigations of *C. tritonis* are warranted to establish gene function, identify candidate settlement cues and explore their possible application in aquaculture and conservation.

**Table 3.** Inducers of metamorphosis and settlement for gastropod veliger larvae (updated and modified [123]). Species for which data is limited to only settlement in presence of live prey (i.e., chemical has not been identified) have been omitted here, but are listed in [123]. Cues tested for *Charonia tritonis* [25] but found to be ineffective are listed as a comparision.

| Species | Compound | Solution/Dose/Time/ % Metamorphosis | Reference |
|---|---|---|---|
| *Concholepas concholepas* | Adult conspecific shells covered in barnacles | Up to 4–5 days, 100% | [124] |
| *Crepidula fornicata* | 20 mM KCl | 50% settlement after 30–50 min | [125] |
| | Adult conspecific conditioned water | 40% settlement after 50 min | [125] |
| | Conspecific pedal mucus | 25% settlement after 50 min | [125] |
| | Raise KCl to 20 mM | 55%, Highest settlement in those fed *Isochrysis* sp. ($4 \times 10^5$ cells/larva/day) | [126] |
| | Elevated KCl above background by 15–20 mM | 50% within 4 h | [127] |
| | Tested serotonin, dopamine and FMRFamide ($10^{-5}$ M/L) | Measured whether larvae go up (serotonin) or down (dopamine, FMRFamide) in the water column | [128] |
| | Dibromomethane (DBM) | 90–100% metamorphosis at 5000 ppm, combined DBM and KCl | [129] |
| | Red algae extract, γ-aminobutyric acid (GABA), Hydrogen peroxide | 70–95% metamorphosis | [130,131] |
| *Aliger gigas* | Nursery habitat sediment, KCl | | [132,133] |
| | Hydrogen peroxide ($H_2O_2$) | 100% at 10 h in 50 μM $H_2O_2$ | [130] |
| | Extract of *Laurencia poiteaui*; Phycoerythrins and related protein conjugants | 88% metamorphosis | [132,134,135] |
| | Bromomethane | 90% at 600 ppm | [136] |
| *Haliotis discus hannai, H. rufescens H. diversicolor, H. asinina* | conc KCl in normal seawater 9 mM | 40% at 19 mM KCl | [136] |
| | $1 \times 10^{-6}$ M (final) GABA | 37–99% | [137] |
| | Whole *Ulva australis* and *U. compressa* and *Amphiroa anceps* and *Corallina officinalis* | 0.05–0.1 g wet wt algae or 1 cm$^2$ of 95% cover rock (CCA) added to 5 mL wells in 4 mL of seawater. CCA best (80%) | [138] |
| | Supplemented KCl | 50% in 5–10 mM KCl (supplemented) | [139] |
| | GABA | 40% $10^{-6}$ M GABA | [139] |
| | KCl, GABA | >40% 20 mM KCl, >75% $10^{-6}$ M GABA | [140] |
| | Biogenic amines | % metamorphosis at $10^{-6}$ M of GABA (98%), L-glutamate (80%), L-glutamine (0%), β-alanine (16–68%) | [141] |
| | GABA, δ-aminovaleric acid (5-AVA), L-glutamic acid, monosodium glutamate (MSG) | $10^{-1}$ mM 5-AVA (62% at 6 h) > $10^{-3}$ mM GABA (58% at 72 h) > 25 mM MSG (50% at 72 h) > $10^{-3}$ mM L-glutamic acid (48% at 72 h). | [142] |
| | $10^{-3}$, $10^{-4}$, $10^{-5}$, $10^{-6}$ M GABA | $10^{-6}$ M GABA at 2 days, 73% | [143] |
| | 5 spp. Benthic diatoms (*Navicula* spp. and *Nitzschia* spp.) | If fed 5 spp., at 2 days 90–94% | [143] |
| *Phestilla sibogae* | Catecholamine precursor L-3,4-dihydroxyphenylalanine (L-DOPA) | 20–50-fold increase in dopamine and 2-fold increase in norepinephrine production in 6–9-day larvae, treated with L-DOPA (0.01 mM for 0.5 h) potentiated the frequency of metamorphosis | [144] |
| *Hermissenda crassicornis* | *Ectopleura crocea* water soluble secretion; GABA, choline, serotonin, glutamate, K$^+$, Cs$^+$. | induces high proportion of metamorphosis | [145] |

**Table 3.** *Cont.*

| Species | Compound | Solution/Dose/Time/ % Metamorphosis | Reference |
|---|---|---|---|
| *Adalaria proxima* | Presence of *Electra pilosa*; peptide with low molecular weight (<500 kDa) | | [146,147] |
| *Charonia tritonis* | Adult conspecific | 5000 L tank with adult conspecific, >10,000 veliger, 0% | [25] |
| | Adult conspecific conditioned water | 45 L tank conditioned for 12 h with adult female conspecific, ~2000 veliger, 12 h, 0% | |
| | Conspecific intracapsular fluid | 6-well plates; 6 veliger per well, 20 µL intracapsular fluid added to each well, 12 h, 0% | |
| | Adult prey | 45 L tank, adult CoTS, ~2000 veliger, 12 h 0% | |
| | Adult prey conditioned water | 45 L tank conditioned for 12 h with adult CoTS, ~2000 veliger, 12 h, 0% | |
| | Adult prey mucous | 6-well plates; 6 veliger per well, 20 µL CoTS mucous added to each well, 12 h, 0% | |
| | Juvenile prey | 500 mL tank, 10× juvenile CoTS, ~2000 veliger, 12 h, 0% | |
| | Environmental cue: crustose coralline algae (CCA) | 6-well plates; 6 veliger per well, CCA chip, 24 h, 0% | |
| | Environmental cue: CCA methanolic extract | 6-well plates; 6 veliger per well, 5, 10 µL mL$^{-1}$, 12 h, 0% | |
| | Sediment (1–1000 µm) from aquaria (live rock, coral, macroalgae, assemblage of other reef organisms) | 500 mL tank, ~2000 veliger, 12 h, 0% | |
| | Filtered (60 µm mesh) sediment from aquaria (live rock, coral, macroalgae, assemblage of other reef organisms) | 500 mL tank, ~2000 veliger, 12 h, 0% | |
| | Multivitamin | 500 mL roller tank, ~2000 veliger, 0.05 multivitamin capsule, 12 h, 0% | |
| | KCl | 500 mL tank, ~2000 veliger, 10, 20 mM, 12 h, 0% | |
| | Synthetic peptides: Serotonin, GLW-amide, WW-amide, APGW-amide, FRMF-amide, sCAP-amide, FF-amide, FF-amide 2, FV-amide, ADRYSFFGGL, Allotropin, Cerebrin, Conopressin, Myomodulin, KPGW-amide, GnRH, Egg laying hormone, Dopamine, L-DOPA | 6-well plates; 6 veliger per well, 10 µM mL$^{-1}$, 12 h, 0% | |

The post-settlement biology, including juvenile growth rates, of some marine gastropods is understood. In newly settled juvenile *Cabestana spengleri* (Tonnoidea: Cymatiidae), growth rates have been estimated at 0.3 mm (shell length) day$^{-1}$ and *C. muricinum* at 0.3–0.4 mm day$^{-1}$ [149,150]. In contrast, the growth rate of recently settled juvenile *G. muricinum, M. aquatilis* and *M. pilearis* (Tonnoidea: Cymatiidae) is much higher, averaging 0.6–0.7 mm day$^{-1}$ for an extended period up to 6 weeks, the highest rates reaching 0.8–0.9 mm day$^{-1}$ [121]. Such high growth rates, coupled with an abundance of food, allow these tritons to achieve formation of the first varix, e.g., within 33 days of settlement for *G. muricinum* and between 50–57 days for *M. aquatilis* and *M. pilearis* [121]. Reaching this life stage is critical for reducing vulnerability to predators [151], but the factors that govern this transition remain a significant knowledge gap in the life cycle of *C. tritonis*. Similarly, adult growth rates are unreported [151].

## 7. Management of Charonia tritonis

*7.1. Threats to Population Recovery*

Throughout their habitat range *Charonia* spp., including *C. tritonis*, are considered uncommon, rare or with seriously depleted populations approaching extirpation [2,57,152,153]. Unregulated harvesting of *C. lampas* and *C. variegata* has severely impacted numbers in the Mediterranean Sea [154–156]. Similarly, in the 1950s *C. tritonis* were regularly observed in the Atlantic and Caribbean, but are now reported to be uncommon to rare throughout [55,157]. Anecdotal evidence indicates that *C. tritonis* were abundant on the GBR prior to incidental collection in the 1930s [6]. Between 1947 and 1960, income from commercial harvesting of Bêche-de-mer (sea cucumber) and *Rochia* spp. (a top-shelled sea snail commonly known as Trochus) in northern Australia was supplemented with *C. tritonis*, with anecdotal records indicating over 800 *C. tritonis* shells were collected from Cooktown to Palm Island in a single trip [8]. Based on these records, an estimated 10,000 *C. tritonis* shells were collected annually [8] and by the 1970s they were considered uncommon [7]. Trade statistics of ornamental shell collections and sales reveal there was a considerable increase in the volume of *C. tritonis* shells traded in the 1970s along with a further depletion of their populations on harvested reefs [9,158,159]. However, as insufficient scientific data exists for harvesting or trade figures, it is difficult to accurately determine whether their rarity today is a result of overexploitation alone.

Between 1966 and 1972, dive surveys conducted on over 130 GBR reefs located only 78 *C. tritonis* [160]. Another study surveying the reefs between Princess Charlotte Bay and the Palm Islands and spanning two years (1966–1968) only found 28 *C. tritonis* [6]. By the late 1980s, as part of the program to cull CoTS on the GBR, 30 divers making 90 dives over 2 weeks were successful in locating only 12 individuals [161]. Furthermore, a population density of <1 *C. tritonis* per km$^2$ was extrapolated based on a 12 month survey (430 h diving time; 1993 to 1994) of 12 GBR reefs between Port Douglas and Airlie Beach [162]. By 2016, divers of the 'Targeted Crown-of-thorns Starfish Control Program' reported sighting, on average, one *C. tritonis* triton per 10 day CoTS culling trip [163]. Similar anecdotal evidence extends to other countries. In the 1960s, local Tongan fisherman regularly collected up to seven *C. tritonis* per day, whereas by 1993, and despite a bounty, none were located over a two-month period [162]. During a 6-month survey of Guam reefs, divers sighted only seven *C. tritonis* [164]. Despite the Thailand Tropical Marine Mollusc Programme [165] listing *C. tritonis* as a 'target' species of interest, only three specimens were procured between 1997–1998. Such anecdotal evidence within the grey literature suggests extreme rarity, and at such low population densities, and as a dioecious species, the probability of encountering a mate and successful reproduction may be severely limited, even after the introduction of marine protected areas [166], as has been the case for the endangered Caribbean Queen conch, *Aliger* (formerly *Strombus*) *gigas* [167,168]. However, it should be noted that the cryptic and nocturnal nature of *C. tritonis* may impede visual counts during daylight hours and result in underestimated population estimates.

Chesher [164] hypothesized that the reduction in the standing stock of *C. tritonis* associated with over-harvesting might have been sufficient at the time for CoTS populations to rise above a critical minimum leading to conditions conducive to outbreaks. Modelling studies have since predicted that in higher numbers *C. tritonis* may suppress CoTS numbers and potentially limit population outbreaks [169–171], yet these models suffer from a lack of verified or adequate information describing the predator-prey dynamics between *C. tritonis* and CoTS [172] and rely on best guestimates of previous and current *C. tritonis* populations.

Collection of *C. tritonis* has been prohibited in Australia since 1983 [173], although illegal poaching has been reported on the GBR. In addition, as a demonstration of latent demand, *C. tritonis* shells have continued to be imported annually into Australia (David Savage, QNPWS pers. comm. in [162]) and more recently traded over the internet [174]. To secure its future, a proposal was submitted in 1993 to include *C. tritonis* on the CITES Appendix II list as a species that may become threatened or extinct unless trade is closely controlled. This proposal was unsuccessful due primarily to the lack of evidence on its biological and trade status, i.e., the

Berne criteria for listing could not be met [175–177]. Regardless, and independent of international agreement, many Indo-Pacific countries have banned the collection or exportation of *C. tritonis*: Australia (1969), India (1972), Seychelles (1969/1978), Fiji (1971), Indonesia (1987) and Philippines (2001). Other countries and jurisdictions, including Guam, Vanuatu and Kenya, have regulated collection [2,9,153–156,178–182], yet several *Charonia* spp. continue to be illegally traded in large volumes [2,183,184]. In countries where collection is not banned, *C. tritonis* are deemed to be locally extinct or extremely rare, i.e., Thailand [57].

With *C. tritonis* now protected on the GBR, the presumption is that populations are slowly returning to pre-1960 levels yet predicting population recovery timeframes is challenging as little is known of the natural pressures they face. In the Caribbean, predation on *C. variegata* by rays (*Aetobatus* sp.) and turtles (*Caretta caretta* and *Eretmochelys imbricata*) has been observed, these predators crushing the shells [55]. Groups of moribund and dead *C. variegata* were also observed; their opercula found near their intact shells with no obvious cause of death, although *Octopus* sp. was ruminated [55]. Living and dead shells have been observed to be badly pitted by bioeroding boring sponges of the genus *Cliona* [55,185], and X-ray has revealed the extent of internal damage sustained by larger (presumably older) specimens [185]. While such infestations have been observed in other gastropods [186] and found to be responsible for extensive shell damage in larger *C. tritonis* shells it is not known whether they cause mortality. Aside from these natural pressures, anthropogenic and environmental stressors present a real and continuing threat to all *C. tritonis* life stages: eggs [187], larval development [188,189], larval diet [190], adult growth [191], shell [192,193] and predator-prey interactions [194]. The cumulative impacts of these, together with the shell still being highly coveted by collectors and the long planktotrophic development phase, may slow recovery to a point where *C. tritonis* numbers will never be naturally restored. A genetic study of *Columbella adansoni* (Family Columbellidae), which also has planktotrophic development, revealed no phylogeographic structure, low interpopulational variance, low genetic diversity and a lack of spatial structure in the distribution of the genetic diversity confirming pelagic larval dispersal to be a critical factor driving high genetic connectivity [195]. Similarly, pelagic larval dispersal was established as the primary factor driving the high level of genetic connectivity in *Talisman scrobilator* (Family Bursidae) over vast distances and throughout its habitat [196]. Unfortunately, no such data is available for *C. tritonis*. Targeted conventional and eDNA surveys, in combination with population connectivity studies and further research into the planktotrophic and juvenile life phases, are essential to assess the gene flow between local and regional *C. tritonis* populations and determine the influence of pelagic larval dispersal. Revealing the pattern of genetic connectivity in *C. tritonis* is highly relevant for its conservation and is particularly important if the population is reliant on the influx of planktotrophic larvae from regions where the shell continues to be overexploited.

### 7.2. Aquaculture and Stock Enhancement Potential

Food security and pharmaceutical biodiscovery has driven the development of aquaculture programs for over 36 marine gastropods (Table 4), including several endangered species [122,197–201]. Some of these programs have also been instrumental in species' conservation. For example, native populations of *A. gigas* have been severely depleted throughout the Caribbean to the point of being threatened and several aquaculture programs to restock these for commercial harvesting have now been established [135,166,199,202–206]. Similarly, wild populations of *Rochia nilotica* in Vanuatu and Vietnam have been successfully replenished through captive breeding [207,208]. More recently, *Ostrea lurida* aquaculture programs have been established along the west coast of North America, with the primary aim being conservation of the species and recovery of locally extinct communities to restore ecosystem functions; harvesting being a secondary (i.e., future) consideration [209].

**Table 4.** Marine mollusks which are produced at commercial scale through aquaculture. Modified from [210,211]. [c] carnivore, [h] herbivore, [d] detritivore, [f] filter feeder, [*] conservation achieved.

| Group | Species |
|---|---|
| **Bivalvia (44)** | |
| Oysters [d/f] | *Ostrea edulis, O. chilensis, O. conchaphila, Magallana gigas, Crassostrea virginica, Saccostrea glomerata* |
| Mussels [d/f] | *Mytilus edulis, M. galloprovincialis, M. chilensis, Perna canaliculus, Anodonta cygnea, Aulacomya atra, Choromytilus chorus, Modiolus* spp. |
| Scallops [d/f] | *Mizuhopecten yessoensis, Aequipecten opercularis, A. (Agropecten) irradians, Argopecten purpuratus, Mimachlamys varia, Pecten maximus* |
| Clams [d/f] | *Mercenaria mercenaria, Corbicula fluminea, Anadara broughtonii, Cyclina sinensis, Venus verrucosa, Donax* spp., *Mya arenaria, Leukoma staminea, Saxidomus gigantea, Tresus nuttallii* |
| Carpet shells [d/f] | *Ruditapes decussatus, Ruditapes philippinarum, Venerupis corrugata, Polititapes rhomboides* |
| Razor clams [d/f] | *Sinomovacula* spp., *Ensis ensis, Panopea abrupta* |
| Cockles [d/f] | *Tegillarca granosa, Cerastoderma edule, Cardiidae* |
| Pen shell clams [d/f] | *Atrina* spp. |
| **Gastropoda (+5)** | |
| Snails | *Rapana* spp. [c], *Babylonia* spp. [d/c], *Buccinum undatum* [c], *Aliger gigas* [d/h,*], *Strombus pugilis* [d/h], *Rochia nilotica* [h,*], *Stromboidea* [h/d] |
| Abalone [h] | *Haliotis rufescens, H. discus, H. tuberculata* |
| **Cephalopoda (1)** | |
| Octopus [c] | *Octopus* spp. |

There are four critical biological stages which require optimization for successful gastropod aquaculture: (1) broodstock procurement, (2) seed (egg and larvae) production, (3) juvenile nursery culture, and (4) sub-adult grow-out to commercial size. However, almost all successful aquaculture programs involve species that are easily collected, spawn year-round, and have lecithotrophic non-feeding larvae that hatch, settle within days, and quickly transition to herbivorous juveniles achieving a minimum market size in 2–3 years [211–216].

Adult *C. tritonis* are rare in the wild, yet they can be collected in sufficient numbers (typically under permit) to establish a broodstock population [25,76,78,217]. Broodstock will breed spontaneously in captivity, with both fertilization and larval hatching readily achieved (Table 2). However, the husbandry requirements of the planktotrophic larvae are substantial, and reliant on suitably nutritional adult and larval diets, the latter typically a cocktail of phyto- and zooplankton [218,219]. Aquaria-reared *C. tritonis* veliger will actively hunt and ingest mixed microalgae, copepod nauplii and adults, artemia nauplii and adult rotifers [25], yet this diet was found to be deficient or suboptimal. Veliger survived for up to 300 days but there was no evidence of larval growth, development or settlement. *C. lampas* veliger fed a diet of diatom and *Artemia salina* survived for 21 days [156]. *C. lampas (sauliae)* fed *Chaetoceros simplex var. calcitrans*, *Isochrysis galbana*, and *Diacronema lutheri* achieved the highest larval survival rate (23% at 15 °C) and shell growth (408 ± 21.52 µm to 625 ± 19.76 µm) over 60 days [220]. In this instance, significant increases in larval survival and growth rates were also achieved when the broodstock (i.e., parental) diet included the preferred starfish prey, *Asterias amurensis* [221]. Marine larvae, including the planktotrophic larvae of many gastropod species [133,135], rely on a hierarchy of environmental sensory cues to locate suitable settlement sites and initiate metamorphosis (Table 3) [222–224]. The critical challenge in rearing *C. tritonis* for stock enhancement remains the identification of these chemical cues (likely mediated by chemosensory receptors [94]) that trigger the cascade of intercellular signaling events and induce metamorphosis and settlement.

Comparative transcriptomic, proteomic and metabolomic (i.e., multi-omics) techniques have proven critical to revealing the underlying molecular mechanisms that regulate hatching, growth, settlement and metamorphosis of commercially and scientifically important molluscan larvae [95], including *Aplysia*, *Biomphalaria*, *Haliotis*, *Cornu*, *Lottia*, *Lymnaea* and *Aliger* [225–228]. Proteomic, transcriptomic and associated expression profiling studies of *A. gigas* have provided significant insight into the reproductive mechanisms and genetic factors that underpin successful spawning in wild populations [226,229]. Furthermore, microsatellite analysis of Caribbean populations separated by 600 km found they were not panmictic (i.e., having limited gene flow) even though the veliger are capable of remaining in the plankton for up to two months and susceptible to environmental factors that promote larval dispersal. Overall, a global deficit of gene heterozygosity was detected, with only four stock populations identified, a finding that led to a reassessment of both local and regional management and conservation efforts [227]. Mitogenomic and transcriptomic resources, based on both larval and adult life stages, have been established for *C. tritonis* and are now revealing information regarding reproduction, and larval development, growth, and competency [37,66,74,94]. However, sequencing the complete nuclear genome and establishing genetic connectivity and diversity of *C. tritonis* is essential for identifying critical stock populations to guide local and/or regional conservation efforts, and to develop species-specific aquaculture methodologies for potential stock enhancement.

## 8. Predator-Prey Dynamics—*Charonia tritonis* and CoTS

### 8.1. Direct Interactions

As pivotal as agents of natural selection, predators drive rapid evolution of key survival behaviors, defensive morphologies and chemical defenses in prey [230,231]. However, despite the various anti-predatory attributes of CoTS (reviewed in [232]) approximately 14 invertebrate and vertebrate species have been *observed* to hunt, attack or consume *live* adult CoTS [4,10,233,234] (Table 5). These predators are mostly generalized feeders and not obligate to CoTS, many preying on injured CoTS or autotomized tissues rather than healthy individuals [10,235].

**Table 5.** Predators observed to prey on healthy live juvenile, sub-adult and adult Crown-of-Thorns starfish. Modified from [4].

| Taxa (Class) | Species | Reference |
|---|---|---|
| Anthozoa | *Stoichactis* sp. *Paracorynactis hoplites* *Pseudocorynactis* sp. | [164,233,236,237] |
| Polychaeta | *Pherecardia striata* | [235,238] |
| Gastropoda | *Charonia tritonis* | [239–242] |
| Malacostraca | *Hymenocera picta* *Tumidodromia dormia* | [235,242–244] |
| Actinopterygii | *Epinephelus lanceolatus* *Lethrinus* spp. *Cheilinus undulatus* *Arothron hispidus*, *A. stellatus* *A. nigropunctatus* *Balistoides viridescens* *Pseudobalistes flavimarginatus* | [10,164,239,241,245–251] |

Although there is limited literature available documenting in situ predation by *C. tritonis* (reviewed by [4]), they have been observed by divers to hunt and feed on CoTS [7,239]. In addition, early field studies reported *C. tritonis* actively seeking and attacking caged CoTS [6]. Preferential predation of CoTS by *C. tritonis* was observed on Grubb and John Brewer reefs

even though *Linckia laevigata* (blue star) was also abundant [161]. Recently, opportunistic surveys in 2020 have photographed *C. tritonis* feeding on adult CoTS (Figure 3).

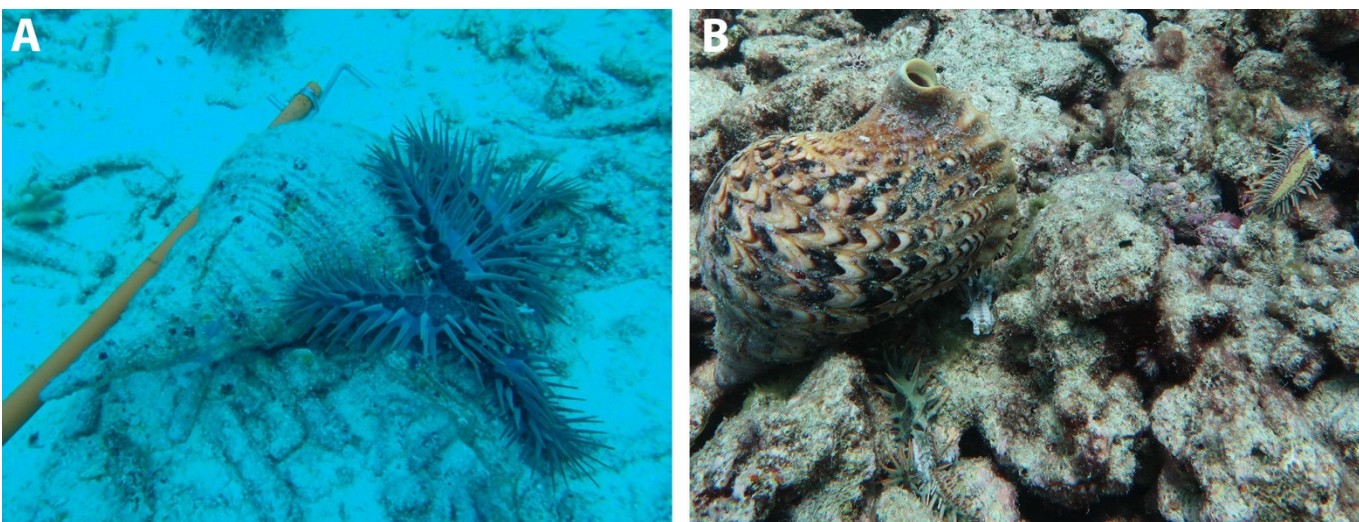

**Figure 3.** Observations of *Charonia tritonis* feeding on Crown-of-Thorns starfish (CoTS). (**A**) Horseshoe Reef, remnants of CoTS limbs evident (Photograph: Kate Osbourne, AIMS Voyage 27, 4 October 2018) and (**B**) Llewellyn Reef, 9 m, ~30 cm shell length [252]; Photograph: crew of Escape GBRMPA Voyage 43, 30 September 2020).

*8.2. Hunting*

Many species of Tonnoidea are specialist predators [33,35]. They are distinct from grazers, having a siphonal canal located within the anterior lip of the shell; the pallial mantle margin folds to fit the lip and directs inhalant water current to a highly developed osphradium containing chemoreceptors capable of detecting prey odors [253]. They also have a long pleurembolic proboscis that has been adapted to deliver, via insertion, toxins and acidic pH 2.0 saliva that are discharged from associated glands [33,254]. Such attacks can cause almost instant immobilization of prey and death, while for some marine gastropods the function of the proboscis is to activate parasitism, as is the case for *G. muricinum* preying on tridacnid clams [121].

*Charonia tritonis* primarily hunts at night. When hunting prey, they randomly sweep their tentacles from side to side. Upon detection of prey odor, this sweeping motion intensifies, and as the odor gradient strengthens, their velocity increases and movement becomes directional [16,37,55]. When in close proximity, the fully extended proboscis (up to 400 mm) perforates the outer skin of the prey targeting its central disc. Rapid paralysis is induced by injection with either a toxin or acidic saliva [55,255,256]. Once paralysed, the prey is held firm by the large muscular foot, after which it is completely enveloped in a thick mucus allowing the radula to rasp the CoTS, rendering the thorny outer skin ineffectual. CoTS secrete toxic hemolytic saponins [257–259] and other toxins into the water column [37] as a chemical defense [260–262] and this mucus likely functions to prevent entry of this toxic cocktail into mantle cavity where it could potentially cause damage to the delicate filaments of the monopectinate ctenidium.

Asterosaponins produced by many species of starfish act aposematically as a signal of unpalatability to potential predatory fish, [263–267], annelids, mollusks, arthropods and vertebrates [257,268–270]. Ultimately, they can prove lethal [262,267,271]. *C. tritonis*, however, are immune to these toxic saponins, readily feeding on live adult CoTS [37]. This immunity may arise from glycosidases. In the liver of *C. lampas*, the glycosidases α-fucosidase, β-xylosidase and β-glucosidase breakdown the asterosaponins by cleaving off the oligosaccharide chain to yield the free non-toxic sterol [272,273]. The sulfur scavenging enzyme arylsulfatase, which is capable of catalysing the breakdown of sulphated saponins,

has also been detected in the salivary glands of *C. tritonis* [37]. This finding supports earlier studies of *C. tritonis* tissues, whereby the principal sterols of CoTS: $\Delta^7$-sterols (34.4% of total sterols), 24-methylcholest-7-enol (15.5%), cholest-7-enol (5.4%), 24-methylcholest-7,22-dienol (6.7%) and acansterol (4.9%) [274], were isolated in significant amounts. Together, these chemical investigations provide indirect evidence for the dietary preference of *C. tritonis*, i.e., asteroids and specifically CoTS.

Not all attacks by *C. tritonis* on CoTS are immediately fatal. CoTS can autotomize the injured arm(s) and later regenerate them [38,64,164], yet observations in captivity have revealed many individuals that initially survive an incomplete *C. tritonis* attack ultimately perish if the proboscis has penetrated the outer skin [185].

*8.3. Prey Preference*

Prey preference is generally explained based on random encounter rates and capture success, with deviations from this indicative of selective predation. For many gastropods, simple encounter rates often fail to account for prey selection indicating they make behavioural diet selections [253,275,276]. For example, *C. (rubicunda) lampas* preys on the most abundant echinoderm within its habitat, but when offered a choice shows a preference for the asteroids *Patiriella regularis* and *Coscinasterias calamaria* [36]. This prey selection is linked to the well-developed chemosensory organs of the Tonnoidea, which enables them to discriminate between prey species and guide hunting. In a controlled aquarium experiment, *C. (rubicunda) lampas* exposed to odor from prey items placed in an upstream current respond with hunting behaviour [36]. Similarly, Y-maze aquarium experiments have shown that waterborne odors released by CoTS attract *C. tritonis* [37,217]. In experimental aquaria, predators are often maintained on a limited diet and may well become impacted by 'ingestive conditioning', hence extrapolation of results from controlled prey preference experiments is fraught with ambiguity [253,277,278]. A further consideration is the ability of predators to undergo dietary switching dependent on the abundance of the preferred prey species. When abundance is high relative to other prey species the number of attacks on and the percent mortality of the preferred prey species is disproportionately large, and disproportionately small when the prey species is relatively rare [279]. Whether this phenomenon influences the feeding preferences of *C. tritonis* remains to be established.

The natural diet of *Charonia* spp. is reported to be predominantly asteroids, followed by holothurians and, to a lesser extent, echinoids [36,55,280,281]. For *C. lampas*, its distribution in the Mediterranean is partially controlled by that of its prey *Holothuria forshali* and *Paracentrotus lividus* [282,283]. In New Zealand, *C. lampas* preferentially preys upon the most common and largest echinoderm in its habitat, *P. regularis* [36]. *C. lampas* presented with meal derived from 15 different species, revealed a preference for asteroid > holothurian > fish > crustacean > other species [284]. A similar preference gradient was observed in *C. lampas* presented with live prey over 30 days: asteroids > holothurian > echinoids, with no predation on mollusks [136]. Based on these findings *C. lampas* was identified as a possible biocontrol agent for the predatory starfish *Asterias amurensis* threatening the security of economically important shellfish fisheries.

Early reports of predation by *C. tritonis* indicated a preference for *Nardoa* sp. [6], although they were also observed to prey on *Stichopus* sp., *L. laevigata* [36] and sub-adult and adult CoTS [7]. This was supported by caged experiments whereby *C. tritonis* was observed preferentially feeding on asteroids other than CoTS if given a choice [239]. Regardless, *C. tritonis* (n = 15) held in a fenced enclosure with 100 adult CoTS over three months consumed 1.5 CoTS week$^{-1}$ [6]. In another study, two *C. tritonis* consumed ten small and three large CoTS month$^{-1}$ [285,286]. Given infested reefs have higher number of CoTS, and that *C. tritonis* would presumably consume whichever echinoderm species they first encounter, these studies suggest they would feed more so on CoTS. Observations of *C. tritonis* feeding preferences in the laboratory confirm they favor asteroids, yet the preferred species appears to vary. When presented with live asteroids (CoTS, *Culcita novaeguineae* and *Echinaster*), echinoid (*Diadema setosum*) and holothurians (*Holothuria atra* and *Stichopus chloronotus*) [80],

a single adult *C. tritonis*, maintained in captivity for 2 years, preferentially hunted and consumed CoTS, while *C. novaeguineae* were only partially consumed. Both *H. atra* and *S. chloronotus* were also readily consumed, however, they induced symptoms of anesthesia in *C. tritonis* post-feeding [80]. When offered both CoTS and *Linckia* in equal numbers, *C. tritonis* attacked and consumed all CoTS entirely within 12 h of being introduced, whereas some *Linckia* remained untouched, suggesting a dietary preference for CoTS [161]. *C. tritonis* broodstock have been successfully maintained on CoTS, *C. novaeguineae*, *H. atra*, *S. chloronotus* [78] and *Stichopus horrens* [76]. Hunting behaviour was initiated 83% of the time for CoTS, 57% for *C. novaeguineae* and 24% for both *H. atra* and *S. chloronotus*. CoTS were completely devoured while the other three species were either very slowly or only partially consumed [78]. *Charonia tritonis*, fed solely on CoTS, have been observed to complete the entire hunt, attack and consumption of an adult CoTS within 4 h to 24 h [37]. Recent citizen science surveys of *C. tritonis* have observed feeding on CoTS, *Linckia* sp. and *C. novaeguineae* [252].

CoTS not only exhibit a strong predator avoidance reaction when in direct physical contact with *C. tritonis* [161], they also display a rapid fleeing response when exposed to *C. tritonis*-conditioned water; CoTS will actively move away from the source of the *C. tritonis* odor [16]. This chemosensory-driven escape response provides further evidence for CoTS as the preferred prey of *C. tritonis* and also supports earlier sightings of *C. tritonis* on CoTS infested reefs, feeding predominately on CoTS [6]. However, to determine the true potential of *C. tritonis* as a CoTS biocontrol agent, there remains a need to establish its full feeding spectrum and prey preference, as well as the 'attractiveness' of these alternate prey species to *C. tritonis*, through both physical presence and choice experiments. Furthermore, to predict the extent of collateral damage to non-CoTS prey populations, the proximity deterrence effects of physical presence (i.e., hunting and consumption) and/or odor (i.e., non-consumptive) of *C. tritonis* on the behaviour and mortality of these species needs to be established.

*8.4. Indirect Interactions*

Prey population density is generally directly mediated by the predator via kill and consumption rates [287]. This process is often referred to as a density-mediated indirect interaction (DMII) and can impact on prey resources as well as other non-prey species [288–298]. In Trinidad, *C. variegata* have been observed hunting in pairs, methodically herding and attacking aggregated spawning *Echinaster sentus*. Individual prey was only partially consumed before the pair resumed hunting and feeding on yet more individuals. This DMII thereby exerted downward pressure on the entire *E. sentus* population [55]). The removal (either unintentional or deliberate) of such marine benthic predators can influence not only the population density of the predominate prey, but also lead to downstream effects on the broader benthic community [299]. The lack of accurate surveys of *C. tritonis* on the GBR pre- and post-exploitation makes it difficult to assess whether this same knock-on effect is a driver for the increased frequency and intensity of CoTS outbreaks.

The mere presence of predators in a community can have significant influence on prey, forcing them to modify their condition (i.e., alter a trait), including phenotype (body shape, armor and size), behaviour (refuge seeking), and physiology (chemical defenses), a process referred to as a trait-mediated indirect interaction (TMII) [300–311], regardless whether or not they consume prey items [312–316]. In essence, while modifying traits minimizes risk of predation, they may inadvertently result in sub-optimal performance of the prey, i.e., slowing and/or delaying growth and maturity [317]. In aquatic ecosystems, non-consumptive TMII effects generally exceed DMII consumptive effects [318,319]. Risk perception by prey, such as that displayed by CoTS in the presence of *C. tritonis*, therefore plays a dominant role in marine trophic interactions, both temporally and spatially, and influences ecosystem stability [319–323].

Quantitative data supporting the ecosystem-wide impacts of TMIIs induced specifically by waterborne predator signals has been reported [231,294,317,324]. These chemically

mediated phenomena, referred to anecdotally as 'landscapes of fear' [312,319,325], are finding application in biocontrol strategies [317,326] including in the marine environment [327]. Regarding CoTS, when exposed to the waterborne chemical odor of *C. tritonis*, representing a short-lived but unpredictable high-risk situation [328–332], they exhibit predator avoidance behaviour [16]. The intensity, persistence [333,334], scale and direction [335,336] of the odor source provides prey with crucial information on immediate risk [337–340] and determines the predator's zone of impact. Furthermore, chemoreceptor sensitivity and specificity is critical if the prey to discriminate such signals [341]. In this context, novel CoTS control technologies are being developed to exploit this predator avoidance behaviour [16,17].

### 8.5. Attributes of a CoTS Biocontrol Agent

Biocontrol is the use of natural enemies to control pest species [342] and has been considered in the context of the marine environment. However, strategies used, especially the choice of predator, the practicality of implementation, and the scale of the effect required are serious issues that need to be overcome to effectively control marine pest species, particularly mobile ones such as CoTS [327,343].

Effective biocontrol agents generally have three attributes: prey specificity, a reproductive rate similar to that of the pest species, and capacity to thrive in the prey's habitat [344]. Similarly, their implementation is usually via three main routes: classical, augmentation and conservation. However, Atalah et al. [327] has warned that "classical biocontrol based on the deliberate introduction of non-indigenous agents has a high risk of leading to adverse non-target effects in marine environments and cannot be justified". Augmentative inundation strategies, which involve the periodic release of a natural enemy without establishing a permanent predator population, are more amenable to marine environments, while conservation strategies, which are more complex to implement, are considered the most acceptable and truly sustainable approach [24]. *C. tritonis*, being an indigenous species and a specialist predator of CoTS in their native range, is suited to the latter two strategies and deserves further investigation. For example, current (laboratory-derived) knowledge indicates the net predatory effect of *C. tritonis* on CoTS is largely due to (non-lethal) TMIIs, yet because of their overexploitation it is not known to what extent they would also contribute to DMII should their population be returned to pre-exploitation numbers, although their status as a primary predator of the adult CoTS (e.g., targeted hunting and lethal consumption) suggests this is likely.

CoTS outbreaks represent a unique problem with respect to their pest status. CoTS are naturally endemic to the Indo-Pacific and play a beneficial role in promoting coral diversity. As such, efforts to control them are not focused on eradication, as is the case in most pest control programs, but rather suppression of populations. Although the drivers of CoTS outbreaks are still highly debated, the release of predation pressure because of over-exploitation of predators [345] means the implementation of augmentative inundation and conservation biocontrol strategies are likely to offer a sustainable approach to suppress CoTS populations in the longer term with the added advantage of restoring *C. tritonis* populations.

Gastropod species have been used, with varying degrees of success, as biocontrol agents (Table 6). Of note are the two predatory marine gastropods *Conus textile* and *Babylonia areolata*. Both have been investigated for their potential as biocontrol agents for *M. pilearis,* a gastropod snail that feeds on the commercially important oyster *Pinctada fucata* [346,347], and found to be effective agents in reducing predation on the oyster. Most recently, the presence of *Vasula deltoidea* has been shown to significantly reduce *Coralliophila galea* corallivory and thus improve *Acropora cervicornis* survival [348]. As for these gastropods, *C. tritonis* possesses many of the biological and ecological traits required for a biocontrol agent (Table 7) and their potential to limit population outbreaks of CoTS is further supported by modelling and tracking studies [72,169–171]. Yet, lessons learnt from unsuccessful campaigns, such as that of *Euglandina rosea* which preferentially hunts

the native *Achatinella lila* over the invasive target pest *Lissachatina fulica* [349], need to be considered for *C. tritonis* to understand the likely impact to non-target endemic echinoderm species and the ecosystem as a whole.

**Table 6.** Gastropods deployed as biocontrol agents. Modified from [350], see references therein. Updated entries are individually referenced. * denotes marine species, ** freshwater species.

| Species | Target of Control; (N) = Unsuccessful, (Y) = Moderate Success, (P) = Potential | Genome of Gastropod | Mitogenome of Gastropod |
|---|---|---|---|
| *Babylonia areolata* * | *Monoplex pilearis* (Y) [346]—mitogenome reported [351] | No | Yes [352] |
| *Conus textile* * | *Monoplex pilearis* (Y) [346] | No | Yes [353] |
| *Edentulina affinis* | *Lissachatina fulica* (N)—mitogenome reported [354] | No | No |
| *Edentulina obesa bulimiformis* | *Lissachatina fulica* (N) | No | No |
| *Euglandina rosea* | *Lissachatina fulica* (N), *Cornu aspersum* (N), *Otala lactea* (N), *Rumina decollata* (N), Slugs (N) | No | No |
| *Tayloria kibweziensis* | *Lissachatina fulica* (N), *Cornu aspersum* (N), *Otala lactea* (N), *Rumina decollata* (N), Slugs (N) | No | No |
| *Tayloria quadrilateralis* | *Lissachatina fulica* (N) | No | No |
| *Gonaxis vulcani* | *Lissachatina fulica* (N) | No | No |
| *Gulella bicolor* | *Lissachatina fulica* (N), *Subulina octona* (N) | No | No |
| *Gulella wahlbergi* | *Lissachatina fulica* (N) | No | No |
| *Marisa cornuarietis* ** | Freshwater weeds and snail vectors of schistosomes; *Biomphalaria glabrata*, *Biomphalaria pfeifferi*, *Bulinus tropicus*, *Bulinus truncatus*, *Hydrilla verticillata*, *Eichhornia crassipes* (Y) | No | Yes [355] |
| *Melanoides tuberculata* ** | *Biomphalaria glabrata*, *Biomphalaria straminea*, *Biomphalaria havanensis*, *Biomphalaria peregrina*, *Biomphalaria helophia* (N) | No | No |
| *Natalina cafra* | *Lissachatina fulica* (N), *Otala lactea* (N), *Rumina decollata* (N), Slugs (N) | *No* | *No* |
| *Oleacina straminea* | *Lissachatina fulica* (N) | No | No |
| *Pomacea glauca* ** | *Biomphalaria glabrata* (N), *Pistia stratiotes* (N) | No; available for *P. canaliculata* | No; available for *P. canaliculata, P. diffusa* & *P. maculata* [356–359] |

**Table 6.** *Cont.*

| Species | Target of Control; (N) = Unsuccessful, (Y) = Moderate Success, (P) = Potential | Genome of Gastropod | Mitogenome of Gastropod |
|---|---|---|---|
| *Ptychotrema walikalense* | *Lissachatina fulica* (N) | No | No |
| *Rumina decollata* | *Cornu aspersum* (N) | No | No |
| *Salasiella sp.* | *Lissachatina fulica* (N) | No | No |
| *Streptaxis contusus* | *Lissachatina fulica* (N) | No | No |
| *Tarebia granifera* ** | *Biomphalaria havanensis, Biomphalaria peregrina, Biomphalaria helophila* (N) | No | No |
| *Vasula deltoidea* * | *Coralliophila galea* (Y), [348], *C. abbreviata* (P) [360] | No | No |

**Table 7.** Summary of characteristics of *Charonia tritonis* and amenability to Crown-of-Thorns Starfish (CoTS) biocontrol. Concept modified from [361]. Prey specificity, predation efficiency, lifespan, and secretion of the 'landscape of fear' feature as key attributes. Amenability is ranked as limited (+), likely (++) and certain (+++).

| Characteristic | Definition | *Charonia tritonis* | Ranking |
|---|---|---|---|
| Narrow host range [362,363] | Generalized predators; preference for the target pest population in the presence of alternate natural prey | Echinoderm specialists; preference for CoTS over other echinoderms not established | ++ |
| Climatic adaptability [364] | Adaptability to the introduced environment, including to environmental extremes | Endemic to GBR | +++ |
| Synchrony with prey life cycle [365] | Should be present when the CoTS juveniles first emerge. | Long-lived; likely decades—unconfirmed | +++ |
| | Self-replicating capacity; High reproductive potential with large numbers of offspring. | lays large clusters of capsules—2000 larvae per capsule | ++ |
| | Population growth rates; teleplanic long-lived oceanic larval phase | Likely slow—unconfirmed | + |
| | More than one generation is completed for each generation of the pest | annual spawner on GBR | ++ |
| | Longevity | Likely decades—unconfirmed | +++ |
| Efficient search ability | Prey detection ability even when prey is scarce | Chemosensory capacity | +++ |
| Short handling time | Higher predator consumption rates equate to greater number of attacks on prey. Small populations of efficient natural enemies may be more effective biocontrol agents than larger populations of less efficient species. Effective biocontrol agents reduce or suppress a pest population below a defined threshold. | Only eat 1–2 CoTS per week | + |
| Survival at low host (prey) density | The type of biocontrol used will depend on several factors for this to be effective | Will prey on other echinoderms in the absence of CoTS | +++ |

## 9. Future Prospects

Presented here is the current state of knowledge regarding the giant triton, *Charonia tritonis*, overfished throughout its Indo-Pacific habitat and now considered rare and endangered. With much of the knowledge limited to their morphology and anatomical biology, conservation efforts have focused on easy-to-implement local protection measures, however, these have been shown to be inconsistent, with some governments imposing a

strict no-take policy, while others none. In addition, population distributions, even within reef systems such as the well-studied GBR, are limited to historical records and recent opportunistic sightings, made even more difficult by their cryptic nature. Hence, there is no way of knowing if *C. tritonis* populations are recovering, static or continuing to decline. Extending traditional biogeographical surveys using molecular-based techniques, supported by a fully sequenced *C. tritonis* genome, will help establish their spatial extent and true numbers, providing a baseline against which future populations can be monitored.

On the GBR, there is no evidence demonstrating the regulation of exploitation (through a no take policy) or the designation of nature reserves (i.e., green 'no-take' zones) has increased *C. tritonis* numbers. Stock enhancement, usually through the introduction of advanced juveniles reared in ex situ breeding programs, can restore populations, and improve the success of conservation outcomes. Yet, attempts to rear *C. tritonis* in captivity have, to date, proven unsuccessful with larval growth and settlement key bottlenecks in the process. Knowledge of their reproductive and early life stage (veliger) biology, particularly the optimal larval diet and the factors that govern settlement, is crucial to overcome these. With little data available regarding the juvenile life stage, discovering when they transition to an echinoderm diet is also imperative (i.e., immediately upon settlement and coinciding with settlement of CoTS, or longer?).

Coral cover on the GBR continues to decline under the pressure of recurring CoTS outbreaks despite significant intervention. To avoid the tipping point beyond which most or all hard corals may disappear, conservation biocontrol, based on the use of an indigenous enemy, represents a promising complementary and sustainable solution to protect coral reefs from CoTS over their full geographical range. With supporting field and laboratory-based evidence suggesting a proclivity for CoTS (i.e., consumptive effect), the *C. tritonis* remains a biocontrol agent of interest. There is also mounting experimental evidence that the chemistry naturally exuded by *C. tritonis* modifies CoTS behaviour [16] (i.e., non-consumptive effect), with the application of such chemistry being considered within the CoTS IPM strategy. However, with predation levels low at 1 CoTS per week, and with the diffusive spread and dilution of chemosensory cues in the aquatic environment reliant on hydrodynamic parameters [366,367], neither trait in isolation is likely to impact on CoTS populations. However, the combination of non-consumptive *and* consumptive effects induced by the presence of *C. tritonis* is likely to have greater success in a proximity deterrence effect on CoTS in situ. This deterrence could promote effective dispersal or prevent aggregations of CoTS at opportune times (i.e., spawning events), thereby indirectly suppressing CoTS populations, especially those under the outbreak threshold. Stock densities of *C. tritonis* needed in such scenarios to elicit the desired affect remain to be determined. These knowledge gaps should be addressed with some urgency.

Mapping of population distributions against those of CoTS, based on knowledge of their predator-prey dynamics, will also establish evidence of the link (or not) with CoTS outbreaks, and if proven so, will improve predictions of future outbreaks and provide another tool (i.e., predator management) for the long-term sustainable control of CoTS populations.

**Author Contributions:** M.R.H. obtained funding. C.A.M. and M.R.H. conducted the literature review and wrote the original draft. C.A.M., M.R.H. and S.F.C. reviewed and edited further versions of the manuscript. All authors have read and agreed to the published version of the manuscript.

**Funding:** This project was funded by the Australian Government National Environmental Science Program Tropical Water Quality (NESP TWQ 2.1.1) Hub, and the Australian Federal Government Department of the Environment and Energy Reef 2050 Sustainability Plan Grant ID: 3600000775. The project was supported by the Australian Institute of Marine Science.

**Institutional Review Board Statement:** Not applicable.

**Informed Consent Statement:** Not applicable.

**Data Availability Statement:** Not applicable.

**Acknowledgments:** We thank Damien Burrows of James Cook University and Frederieke Kroon of AIMS for valuable input and suggestions to the NESP TWQ 2.1.1 report on which this work is based. We acknowledge the Wulgurukaba and Bindal people as the Traditional Owners of the land upon which this review was conducted.

**Conflicts of Interest:** The authors declare that the research was conducted in the absence of any commercial or financial relationships that could be construed as a potential conflict of interest.

## Abbreviations

| | |
|---|---|
| CoTS | Crown-of-Thorns starfish |
| dpf | days post-fertilization |
| eDNA | environmental Deoxyribonucleic acid |
| GBR | Great Barrier Reef |
| IPM | integrated pest management |

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
