# Peer review of "A Review of the Giant Triton (Charonia tritonis), from Exploitation to Coral Reef Protector?"

_diversity, doi:10.3390/d14110961_

Round 1
Reviewer 1 Report
Title
· “Coral Reef Saviour” is rather hyperbolic.
Abstract:
· Syrinx aruanus is the largest shelled marine gastropod.
Taxonomy and distinctive characteristics
· The most recent work dealing with the higher order relationships of Caenogastropoda is Ponder, Lindberg & Ponder 2019. The Biology and Evolution of the Mollusca.
· In your review on taxonomy and systematics of tonnoideans you are missing the following important contribution, which revised the family classification of the superfamily and elevated Charoniidae to a separate family:
o Strong E.E., Puillandre N., Beu A.G., Castelin M. & Bouchet P. (2019). Frogs and tuns and tritons – A molecular phylogeny and revised family classification of the predatory gastropod superfamily Tonnoidea (Caenogastropoda). Molecular Phylogenetics and Evolution. 130: 18-34., available online at https://doi.org/10.1016/j.ympev.2018.09.016
· Line 86: The Taenioglossa is not a formal taxonomic grouping.
· Line 90: Bouchet et al 2005 has been superseded by Bouchet et al 2017: Bouchet P., Rocroi J.P., Hausdorf B., Kaim A., Kano Y., Nützel A., Parkhaev P., Schrödl M. & Strong E.E. (2017). Revised classification, nomenclator and typification of gastropod and monoplacophoran families. Malacologia. 61(1-2): 1-526.
Reproduction:
· Comparative remarks on Ranellidae will need to be revised given the new family classification of Strong et al. (2019).
· Lines 221-222: “A major innovation in the Caenogastropoda is internal fertilization enabling the production of encapsulated eggs thereby providing a protected environment during early trochophore development.” Citation?
· Line 250: “but, unlike other mollusks, there are no nurse cells”: not all mollusks have nurse cells.
· Lines 267-270: “Even though planktonic protoconch I veliger have been successfully maintained (Figure 2C) for up to 300 days in aquaria and shedding of their velar cilia was observed, development to the protoconch II stage did not proceed, with the left tentacle still absent”: This information is repeated in the next section on lines 310-313.
Juvenile growth, development and morphological relationships
· Again, comparative remarks on Ranellidae will need to be revised given the new family classification.
· Line 332: Not all Cymatium feed on Tridacna.
· Table 3 is superfluous given what little analysis is provided of its contents in the text.
Management of Charonia tritonis
· Line 371: “local extinction” = extirpation
· Line 465: The current placement of Bursa scrobilator is in the genus Talisman
https://molluscabase.org/aphia.php?p=taxdetails&id=1472101
Please refer to the World Register of Marine Species (which draws on the classification in MolluscaBase for the mollusks) to confirm the current combinations for all the taxa you mention in the paper.
· Lines 521-527: This information about settlement cues is repeated yet again. The authors need to do a careful reading of the manuscript and eliminate this kind of redundancy and repetition.
Predator-prey dynamics – Charonia tritonis and Acanthaster planci
· Line 578: “Ranellidae”
· Line 598: “hypobranchial chamber” = mantle cavity
· Line 628: C. rubinrnda
· Line 633: “Ranellidae”
· Table 6. This table has some significant omissions, notably Euglandina rosea, a biocontrol agent that went terribly wrong. I do not see the value of providing an edited list, which is then so superficially discussed in the text, notably glossing over the cases of biocontrol failures. Many of these examples relate to freshwater and are not terribly relevant to the discussion at hand.
Reviewer 2 Report
Review A Review of the Giant Triton (Charonia tritonis), from Endangered Species to Coral Reef Saviour?
Abstract and Introduction section
Please, place (Charoniidae) after Charonia tritons,
L291-293. Please add a referenced example for this. “In addition, complementary studies to discover the genes and elucidate the molecular mechanisms underlying oogenesis and embryogenesis are critical to assess viability of eggs and embryos and to understand the drivers of early-life stage mortality”. There are intrinsic and extrinsic factors that are associated to this.
Comment: This is a well-documented review addressing the endangered position of Charonia tritonis and their putative use in biocontrol strategies for Crown-of-Thorns starfish (CoTS). I enjoyed revising the Ms. Apart of the above comments, I do not have more queries.
